| **Open Peer Review** | Genomics and Proteomics | Methods and Protocols

# Improving gene set enrichment analysis (GSEA) by using regulation directionality

Biwen Wang,[1] Frans van der Kloet,[1] Mariah B. M. J. Kes,[2] Joen Luirink,[2] Leendert W. Hamoen[1]

**ABSTRACT** To infer the biological meaning from transcriptome data, it is useful to focus on genes that are regulated by the same regulator, i.e., regulons. Unfortunately, current gene set enrichment analysis (GSEA) tools do not consider whether a gene is activated or repressed by a regulator. This distinction is crucial when analyzing regulons since a regulator can work as an activator of certain genes and as a repressor of other genes, yet both sets of genes belong to the same regulon. Therefore, simply averaging expression differences of the genes of such a regulon will not properly reflect the activity of the regulator. What makes it more complicated is the fact that many genes are regulated by different transcription factors, and current transcriptome analysis tools are unable to indicate which regulator is most likely responsible for the observed expression difference of a gene. To address these challenges, we developed the gene set enrichment analysis program GINtool. Additional features of GINtool are novel graphical representations to facilitate the visualization of gene set analyses of transcriptome data, the possibility to include functional categories as gene sets for analysis, and the option to analyze expression differences within operons, which is useful when analyzing prokaryotic transcriptome and also proteome data.

**IMPORTANCE** Measuring the activity of all genes in cells is a common way to elucidate the function and regulation of genes. These transcriptome analyses produce large amounts of data since genomes contain thousands of genes. The analysis of these large data sets is challenging. Therefore, we developed a new software tool called GINtool that can facilitate the analysis of transcriptome data by using prior knowledge of gene sets controlled by the same regulator, the so-called regulons. An important novelty of GINtool is that it can take into account the directionality of gene regulation in these analyses, i.e., whether a gene is activated or repressed, which is crucial to assess whether a regulon or functional category is affected. GINtool also includes new graphical methods to facilitate the visual inspection of regulation events in transcriptome data sets. These and additional analysis methods included in GINtool make it a powerful software tool to analyze transcriptome data.

**KEYWORDS** transcriptome analysis, GSEA, regulons, functional categories, *Bacillus subtilis*, operons

Genome-wide transcriptome analysis is a commonly used method to measure changes in gene regulation. However, due to the large number of genes measured, data interpretation requires focusing on a limited set of genes. Generally, this is accomplished by setting a threshold level for the expression change of an individual gene and its statistical significance, often expressed as *P*-value. However, this may result in the loss of important information since even a modest up- or downregulation of a gene can have biological implications. Therefore, it can be more informative to look at sets of genes to obtain genome-wide regulatory information (1). Different computational

Address correspondence to Leendert W. Hamoen, l.w.hamoen@uva.nl, or Frans van der Kloet, f.m.vanderkloet@uva.nl.

Biwen Wang and Frans van der Kloet contributed equally to this article. Author order was determined by drawing straws.

The authors declare no conflict of interest.

See the funding table on p. 11.

gene set enrichment analysis (GSEA) tools, such as GSEA (1), DAVID (2), Enrichr (3), and FUNAGE-Pro (4), use annotation databases like Gene Ontology, Kyoto Encyclopedia of Genes and Genomes (KEGG), and Wikipathways (5–7) to link transcriptional changes to biochemical pathways and functional categories. When sufficient knowledge of gene regulation networks is present, gene set enrichment analysis can provide insights into the activity of regulation networks (4). However, current GSEA tools do not take into account whether a gene is activated or repressed by a regulator, and this omission can result in unreliable conclusions.

To determine whether a regulon is activated or repressed, gene set enrichment tools will average the expression differences of all genes of a regulon. However, many transcription factors function as an activator of some genes and as a repressor of other genes, yet both sets of genes are part of the same regulon. As a consequence, simply averaging the fold change of regulon genes will not reflect the activity state of the regulator. For example, in the bacterial model system *Bacillus subtilis,* the response regulator Spo0A regulates 143 genes of which approximately half is by activation and the other half is by repression (8). As a consequence, the average fold change of the Spo0A regulon will not properly reflect the activity of Spo0A. To overcome this problem, it is necessary to take into account the directionality of regulation, i.e., whether a regulator functions as an activator or as a repressor of a specific gene. Therefore, we have developed the gene set enrichment analysis software GINtool. GINtool provides a more accurate activity indicator for regulons and can provide a measure of probability whether a regulator is responsible for the observed transcriptional effect. This is especially useful when dealing with genes that are controlled by multiple regulators. GINtool also includes novel graphical outputs to facilitate the visual analysis of gene set changes in transcriptome data and includes the option to use functional categories for gene set enrichment analyses. Another feature added to GINtool is the option to inspect gene regulation within operons. This can be useful when analyzing prokaryotic transcriptome data since genes within an operon can show unrelated expression differences, indicative of an unknown type of regulation or a wrongly annotated operon. The software package is provided as an Excel plugin to offer a large degree of freedom to analyze and present the data using standard Excel or other graphical tools. Here, we illustrate the different capabilities of GINtool by analyzing transcriptome data from *B. subtilis*.

## RESULTS AND DISCUSSION

### Improving gene set visualization using median absolute deviation of the fold change

To illustrate the suitability of GINtool, we used transcriptome data from an experiment examining the effect of XynA overexpression in *B. subtilis*, an industrial relevant xylanase (9). Experimental details are described in the Materials and Methods. *B. subtilis* is one of the best-studied bacteria, and so far, approximately 220 regulons have been characterized for this organism (8, 10, 11). This information can be collected from Subtiwiki, the main knowledge repository for *B. subtilis*, which contains up to date and easy-to-use tables listing gene functions, regulons, functional categories, operon structures, and other information (8). These tables can be uploaded into GINtool, as described extensively in the manual provided with the GINtool starter package and included in the supplementary information. To upload the transcriptome data in GINtool, the data table should contain gene names and/or unique identifiers (UIs), differential expression values and related statistical values like *P*-values.

First, we tried to find a gene set output format that is visually more easily interpretable than the currently used heat or treemaps [e.g., (12, 13)]. The pitfalls of using heat maps are the limitation of color representation and information overload due to too many details. Treemaps are heatmaps that display also the regulon size, i.e., the number of genes of a regulon, by using rectangles of different sizes [e.g., (14, 15)]. Treemaps use size and color to present data, which can make it difficult to see how small regulons are affected or how uniform the expression differences are within a regulon, which can

be an indicator of the robustness of regulation. To capture the latter, we calculated the median absolute deviation (MAD) of the fold change values of all genes in a regulon (16). It appeared that these MAD values provide a useful means to visualize transcriptional differences between regulons when plotted against the average fold change of regulons (Fig. 1A).

In Fig. 1A, a bubble plot layout was chosen so that the size of a regulon can be displayed as well. The large bubble in Fig. 1A represents the housekeeping sigma factor (SigA) regulon. To give an estimate of the statistical significance of regulon regulation, expressed as *P*-values, GINtool uses the fast gene set enrichment analysis method (1, 17). The effect of grouping genes was taken into account by using the *P*-values adjusted for the false discovery rate. A gene set with genes that cluster, i.e., genes that have a similar fold change, will have a lower *P*-value than a gene set with genes that have a less uniform response. GINtool sorts regulons based on *P*-values and divides them into five categories, the 10% regulons with the lowest *P*-values, then the 10%–20%, 20%–33%, and 33%–50% fractions, and finally the rest (50%) of the regulons with the highest *P*-values. This is indicated in the graph by decreasing color intensities and enables Excel to easily remove less relevant regulons, those with high *P*-values, to make a graph clearer, as is illustrated in Fig. 1B, where 80% of the regulons with the highest *P*-values are not shown. In this case, we also added the names of regulons. These labels can be easily added, moved, or removed in Excel. It is also easy to change from bubble plots to other types of graphs.

## Using directionality of regulation to improve regulator ranking

The average fold change in Fig. 1 is calculated by simply averaging the fold change of all genes of a regulon. However, this can result in a fold change value that is not representative of the regulator, since a regulator can simultaneously function as an activator of some genes and as a repressor of other genes, as outlined in the introduction. Therefore, it is important to take the directionality of regulation into account, i.e., whether a gene is activated or repressed by a regulator. In the case of *B. subtilis*, this information can be found in the regulon file from the Subtiwiki website. Unfortunately, the Subtiwiki regulon file has annotated the directionality of regulation by different qualifiers, such as "activation," "antitermination," "sigma factors," "negative regulation," and others. In GINtool, these qualifiers can be manually divided into either

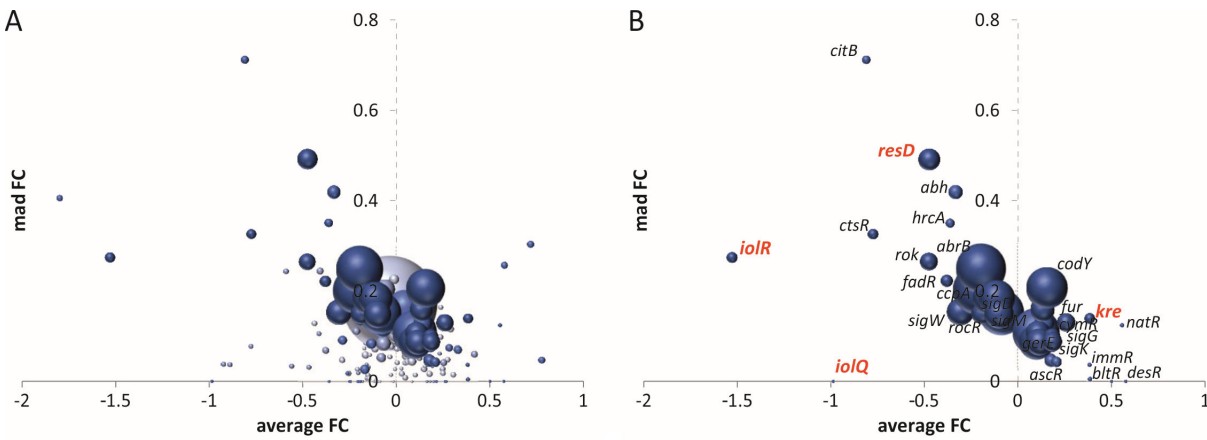

**FIG 1** GINtool visualization of regulon data using the MAD of fold changes. (A and B) Regulon bubble plots showing average fold change (FC, log2 scale) of regulons plotted against the MAD of the fold change (mad FC, log2 scale). The size of bubbles corresponds to the number of genes in a regulon. The color intensity of the bubbles indicates the *P*-values of regulons and goes from dark to light based on the ranking of regulons from lowest to highest *P*-values. The ranked regulons are sorted into five fractions: the 10% with the lowest *P*-values, then the next 10%–20%, 20%–33%, and 33%–50% fraction, and finally the rest (50%) of the regulons with the highest *P*-values. Panel (A) shows information of all regulons, while panel (B) shows only the 20% of the regulons with the lowest *P*-values. The regulons mentioned in the main text are highlighted in red.

the qualifier "downregulated" or "upregulated." Of note, sigma factors are considered positive regulators, and therefore, we considered this type of regulation as "upregulated."

In order to determine whether the regulator of a regulon is activated or repressed, GINtool divides the genes of a regulon into two groups based on whether their fold change direction (down/upregulation) fits with either an "activated" or "repressed" regulator. For example, when Subtiwiki indicates that "Regulator A" is a repressor of "*gene b*," and this gene is downregulated in the transcriptome data, then *gene b* will belong to the group of "Regulon A" genes that correspond to an activated regulator A. GINtool considers the group with the most genes the most relevant regulation situation and will then calculate the average fold change, MAD, and *P*-values for only the genes of this group. When the activated regulator group and repressed regulator group comprise the same number of genes, GINtool will choose the group with the highest average fold change. The result of such an analysis using our transcriptome data is shown in Fig. 2. Because of the selection of only those genes that correspond to the most likely regulator activity, the average fold change, MAD, and *P*-values, and size of the bubbles can differ between Fig. 1B and 2. This is, e.g., illustrated by the higher average fold change difference and lower MAD value of the ResD regulon. What is also noticeable is that the direction (positive/negative sign) of the fold change differs for certain regulons. For example, the IolR and IolQ regulons show a negative fold change in Fig. 1B, but a positive fold change in Fig. 2. The reason for this is that both IolR and IolQ function as repressors, so a downregulation of the genes they regulate, indicated in Fig. 1B, means that both regulators are active, therefore GINtool assigns them a positive value in Fig. 2. This assignment of positive and negative signs is crucial when dealing with regulons for which the average fold change is based on both up- and downregulated genes, such as the transcription factor Spo0A mentioned in the introduction.

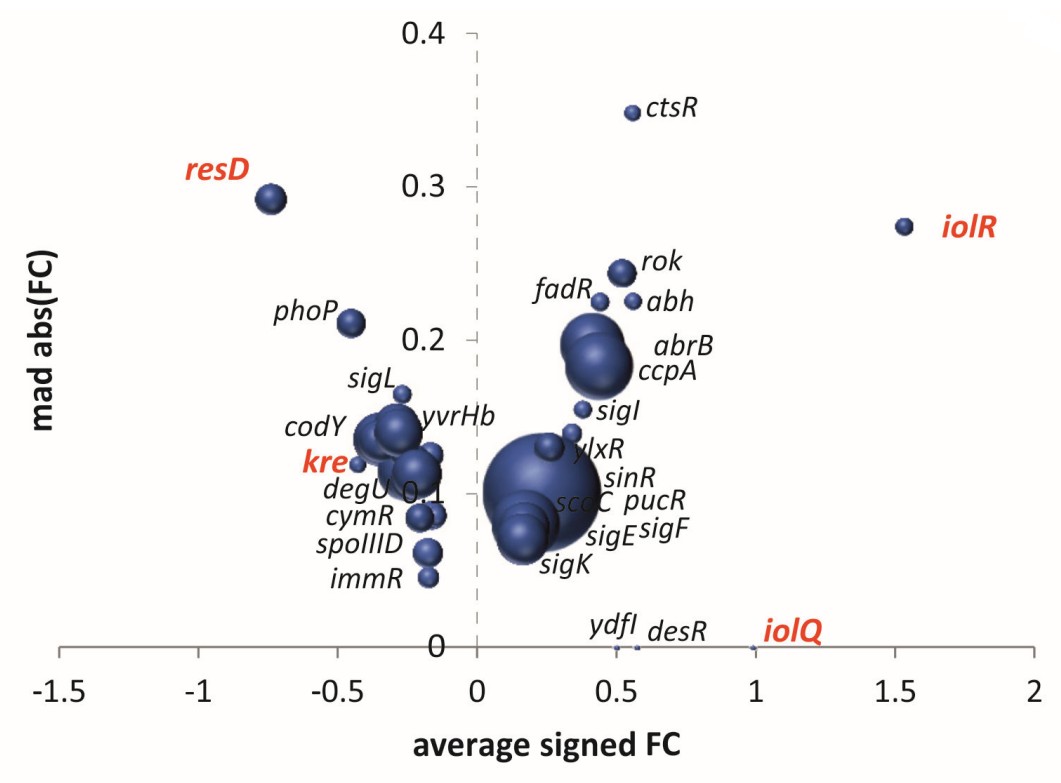

**FIG 2** Regulon plots corrected for the directionality of regulation. Negative and positive average fold change values indicate inhibition and activation of a regulator, respectively. As a consequence, regulons like IolR, IolQ, and Kre (marked in red) switch positions, and ResD (marked in red) has a higher average fold change and lower MAD value in Fig. 2 compared to Fig. 1B. For clarity, only the 20% lowest *P*-value regulons are selected and shown.

From the bubble plots of Fig. 1 and 2, it can still be difficult to discern small regulons when their average fold change and MAD values are small. Therefore, GINtool also provides clearly organized tables, listing the calculated average fold change, MAD, and *P*-values for all regulons under the different regulation scenarios (see manual), enabling customization of data analyses and presentations.

### Ranking regulons using the fraction of regulated genes

Figure 2 shows, by means of bubble size, the number of genes that are in accordance with the most likely activity (activation/repression) of a regulator. However, this number does not indicate what fraction of regulon genes this is, yet this is important to know. For example, when 51% of the genes of a regulon show a regulation that corresponds to an activated regulator, then this regulation is not very relevant, considering that 49% of the regulon genes show an activity that corresponds to a repressed regulator. Yet, for large regulons, even 51% will still give a considerable bubble size. GINtool can work around this problem by calculating the fraction of regulon genes that are used to calculate the best regulator activity. These percentages can then be used to plot the average fold change, as is shown in Fig. 3. For example, this graph strongly suggests that the activity of the regulator Kre is slightly repressed since the activity of at least 90% of its regulon genes correspond to this activity. Of note, also in Fig. 3, the average fold change and *P*-values are only based on those genes representing the indicated fractions, i.e., only those genes that show a differential expression corresponding with the most likely activity of the regulator.

GINtool will list these fraction values in tables (see manual) as well, to facilitate analyses and alternative presentations of the data. One such alternative, which is already incorporated in GINtool, is the option to display the average fold changes against *P*-values in a volcano plot, as shown in Fig. 4. This time, colors indicate the reliability

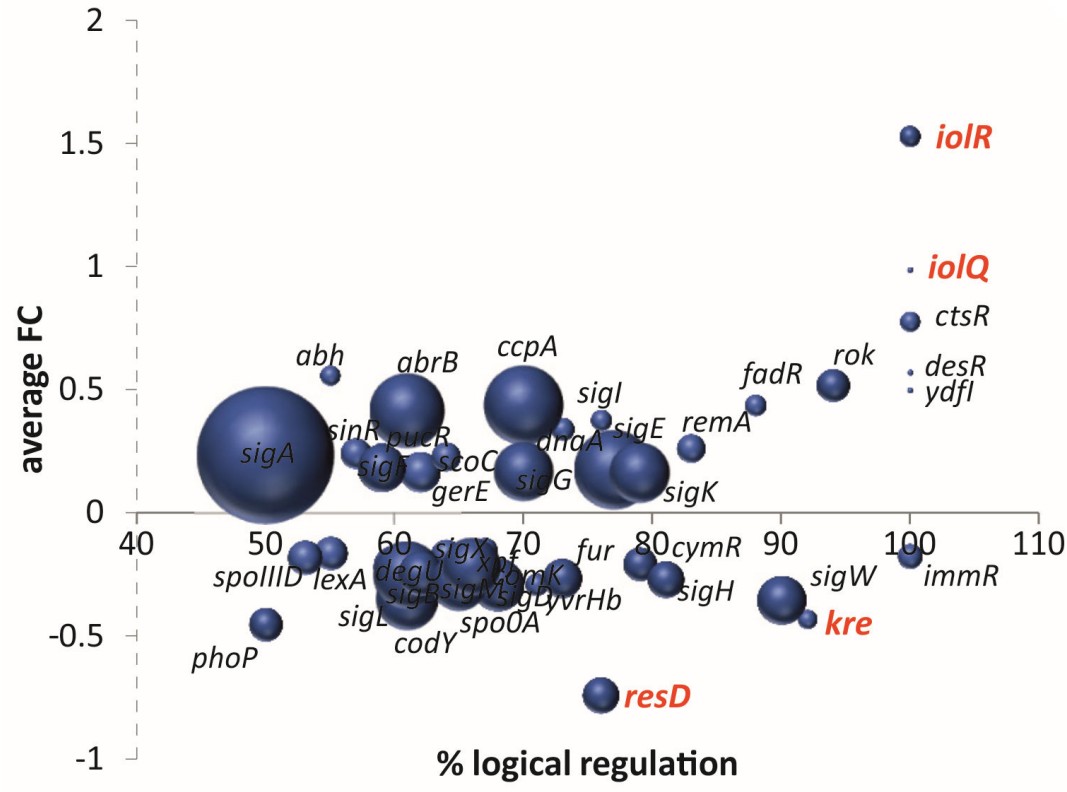

**FIG 3** Ranking based on the fraction of regulated genes. Average fold change from Fig. 2 plotted against the fraction of regulon genes that represent the most likely (% logical regulation) activity of the regulator. For clarity, only the 20% lowest *P*-value regulons are selected and shown.

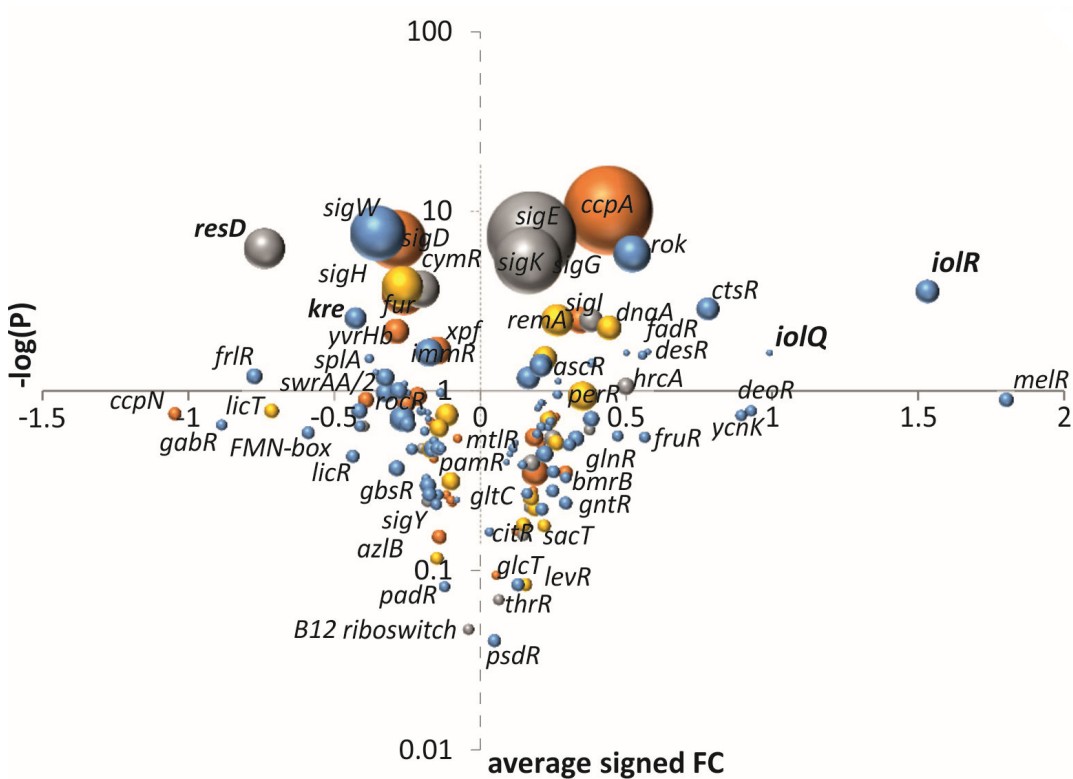

**FIG 4** Volcano plot. *P*-values of regulons are plotted against their average fold changes resulting in a volcano plot. The colors correspond to the reliability of the predicted activity using the percentages shown in Fig. 3. The colors blue, yellow, gray, and orange correspond to regulons of which 100%–90%, 90%–80%, 80%–75%, and 75%–67%, respectively, of their genes show a differential expression corresponding to the same regulator activity. The regulons with percentages < 67% are not selected and not shown here. For clarity, the Y-axis is shown in log scale.

of regulation based on the percentages shown in Fig. 3. For this, GINtool divides regulons into five color categories: regulons of which 100%–90%, 90%–80%, 80%–75%, 75%–67%, and 67%–50% of their genes show a differential expression corresponding to the same regulator activity. Since Excel can selectively show these categories, it makes it easier to show regulons that are most likely affected in the transcriptome data.

## Providing regulon information for every gene

Generally, a transcriptome study will include a table listing of the most strongly up- and downregulated genes. It can be informative to also indicate the regulator(s) that might be responsible for the observed transcription effects. GINtool will show the information of Fig. 3, i.e., the average fold change and the fraction of regulon genes that show the most likely regulation for every gene in a table. A screenshot is presented in Fig. 5. For example, the fifth gene in the list, *iolA*, belongs to three regulons: IolR, CcpA, and SigA. However, the average fold change of the IolR regulon is 1.53, clearly the highest, and all (100%) genes of the IolR regulon follow a fold change direction corresponding to increased IolR activity. Therefore, it is likely that the IolR activity is responsible for the expression difference of *iolA*. Note that the average fold change is a positive value for IolR, whereas the expression of *iolA* is downregulated −2.03-fold. As mentioned before, this is because IolR is a repressor. When this repressor is activated, it will repress *iolA* expression, resulting in the observed downregulation. GINtool will show fold change and fraction information for all the regulons to which a gene belongs but leaves it to the user to decide which regulator is most likely responsible for the observed expression difference.

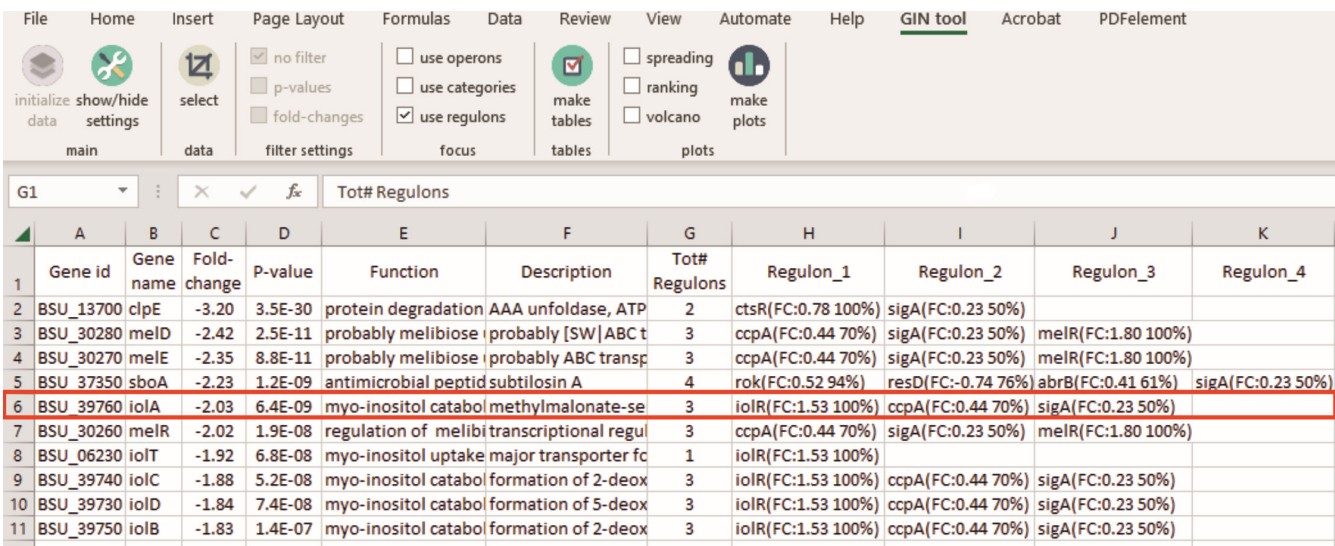

**FIG 5** Linking regulators to genes. Example of a GINtool output table showing the main transcriptome information of genes with the regulons to which they belong. Column G lists the number of regulons to which a gene belongs. Columns H, I, etc., list these regulons with their average fold change (FC) and most likely activity indicated by a negative FC for repression and positive FC for activation. The fraction (%) of genes of the regulon that corresponds to the most likely activity is indicated as well. Of note, the average FC is based only on these genes. The red box highlights the IolA regulon discussed in the main text.

## Use of functional categories

*B. subtilis* is one of the best-studied bacteria, but for most bacterial species, gene regulation networks are not so well characterized. In these cases, the use of functional categories is a good alternative for gene set enrichment analysis. For *B. subtilis* more than 350 functional categories and subcategories have been defined. GINtool can also rank and display the average fold change, MAD, and *P*-values of functional categories in bubble plots, as is done for regulons. The software contains an option to select which level of categories needs to be used, i.e., main categories, subcategories, and/or sub-subcategories etc., as outlined in the manual.

Whether a functional category is up- or downregulated is calculated by averaging the fold change of the genes within a category. Therefore, it can be useful to take into account what fraction of genes in a functional category is up- and what fraction is downregulated, since this ratio can determine the relevance of a functional category for the transcriptome analysis, much like the situation with regulons. GINtool can perform such analysis and display the outcome as bubble plots, like those shown for regulons in Fig. 2 to 4, and in tables. Examples are described in the GINtool manual.

## Visual ranking using spread plots

GINtool can also plot the fold changes of all genes of a regulon or functional category and rank them according to their average fold change, which is often used to show gene set changes. In this case, regulation directionality is not taken into account. Figure 6 shows the result of such a spread plot using the *B. subtilis* regulon information. An example of such a spread plot using functional categories is shown in the GINtool manual. To facilitate visualization of the most important regulons or functional categories, GINtool has the option to show only the most strongly affected gene sets based on average fold change (Fig. 6, inset).

## Analyzing expression differences within operons

Bacteria regulate multiple genes simultaneously by locating them in an operon. However, when arbitrary fold change and *P*-value cut-offs are used to select the most relevant genes, it might appear as if only a few genes of an operon are significantly

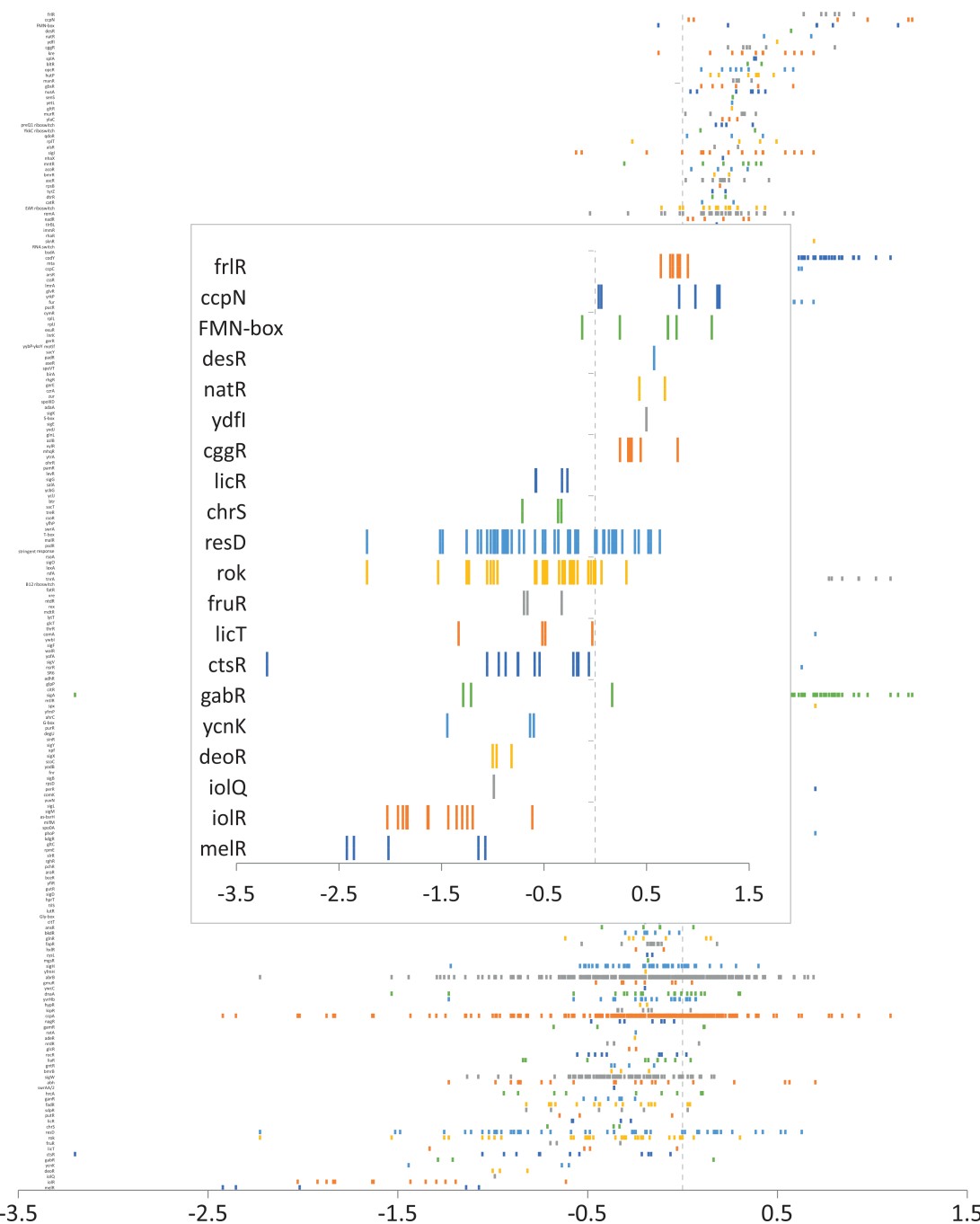

**FIG 6** Ranking of regulons using a spread plot. Example of the ranking of *B. subtilis* regulons based on the average fold change in a spread plot by GINtool. The X-axis scale shows fold change values in log2. Each small bar represents the fold change value of a gene, and each row represents all genes of a regulon. The inset shows the option to display only the most up- and downregulated regulons, in this case, the top 20.

regulated. GINtool has the option to visually examine expression differences of genes in operons. This can also be useful to detect unexpected regulation events. Figure 7 shows a screenshot of such analysis. Information on operon composition was derived from the Subtiwiki database. Interestingly, there appears to be an anomaly for the *yxaJ* operon, since the first gene of the operon is upregulated, whereas the second gene is clearly downregulated. This suggests that the second gene in the operon (*yxaL*) is regulated by an unknown regulator and that this operon is not appropriately annotated. Indeed, an extensive transcriptome study (18) suggests that *yxaJ* and *yxaL* are, at least partially,

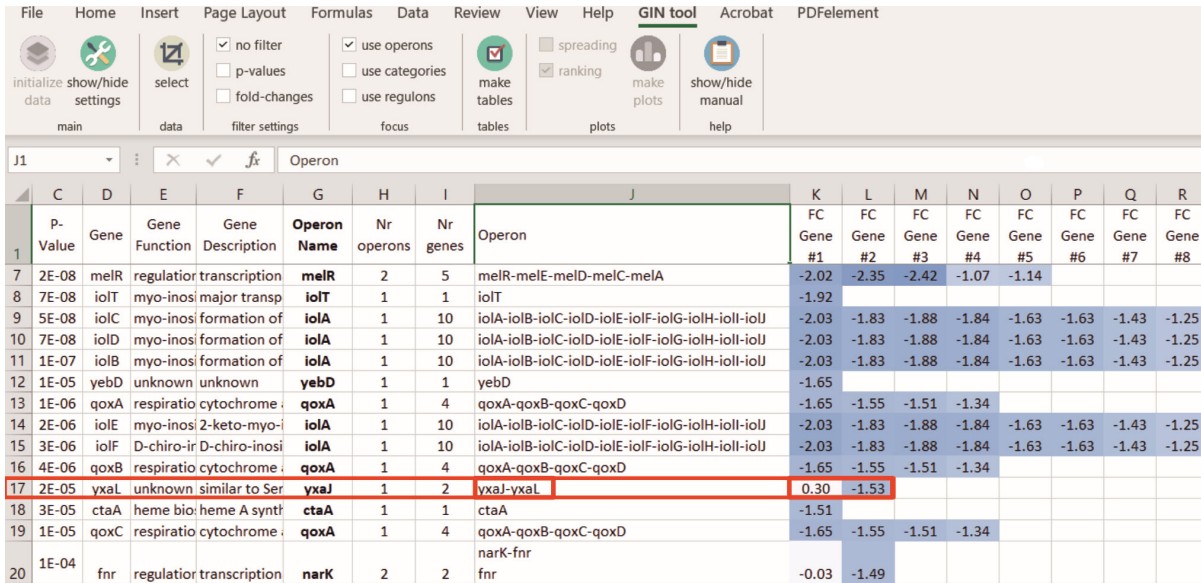

**FIG 7** Expression difference analysis within operons. Example of a GINtool output table showing the main transcriptome information of genes, including the operon(s) the gene belongs to. Column H indicates the number of operons a gene belongs to, column J shows the gene context of these operons, and the rest of the columns (K, L, etc.) show the expression fold change of these genes. Fold changes were color coded using the conditional formatting function of Excel. The red box shows an example of an operon (*yxaJ-yxaL*) with an illogical fold change distribution.

regulated in different ways. In fact, there is a 101-bp spacer region between both genes, thus ample space for a promoter to regulate *yxaL*.

## Conclusion

With the possibility to include gene regulation directionality information, GINtool adds an important dimension to gene set enrichment analyses. Although the software tool has been developed with bacterial transcriptome analyses in mind, the input and output format is flexible so that it can also be used to analyze other omics data sets, including proteomics, making it a broadly applicable tool.

## MATERIALS AND METHODS

### Bacterial strains and general growth conditions

Bacterial strains and plasmids used in this study are listed in Table S1. Nutrient lysogeny broth medium (LB, containing 10 g/L of tryptone, 5 g/L of yeast extract, and 10 g/L of NaCl) was used for the general growth of both *B. subtilis* and *Escherichia coli*. Supplements were added as required: kanamycin (50 µg/mL), spectinomycin (150 µg/mL), ampicillin (100 µg/mL), and isopropyl β-D-1-thiogalactopyranoside (IPTG) (1 mM). For *B. subtilis* DNA transformation, the Spizizen-plus and -starvation media [SMM, containing 15 mM of $(NH_4)_2SO_4$, 80 mM of $K_2HPO_4$, 44 mM of $KH_2PO_4$, 3 mM of tri-sodium citrate, 0.5% glucose, 6 mM of $MgSO_4$, 0.2 mg/mL of tryptophan, 0.02% casamino acids, and 0.00011% ferric ammonium citrate $(NH_4)_5Fe(C_6H_4O_7)_2$] were used, and transformants were selected on LB-agar plates with antibiotic selection (19).

### Plasmid and strain construction

The effect of xylanase overproduction on the transcriptome was tested in *B. subtilis* strain BWB09 that lacked the native xylanase-expressing gene *xynA* and amylase gene *amyE*. These genes were removed from the tryptophan-prototrophic (trp+) wild-type *B. subtilis* strain BSB1 (18) by means of a marker-free clean-deletion procedure (20). Briefly,

purified DNA amplicons of *xynA*-upstream (primer pair BW45 and BW46), *xynA*-downstream (BW41 and BW44), *Sp^R-mazF* cassette (BW05 and BW06), and *xynA* (BW47 and BW48) were fused by overlap PCR (BW45 and BW44) to make the recombinant DNA construct. All primers used are listed in Table S2. Subsequently, the construct was transformed directly into competent BSB1 cells, and transformants were selected on spectinomycin-selective plates. Next, colonies were inoculated in LB liquid containing 1 mM of IPTG, which induced the MazF toxin expression, resulting in the excision of the deletion cassette. Cells were again spread on LB agar to obtain single colonies. Cells from the edge of single colonies were restreaked on LB and spectinomycin plates, and those that grew on LB agar plates, but not on LB spectinomycin plates, were expected to have the *Sp^R-mazF* cassette and *xynA* removed. The marker-less deletion was verified by PCR with *xynA*-internal and *xynA*-external primers (BW42 and BW43). Subsequently, the *amyE* gene was deleted using the same clean-knockout method, eventually resulting in strain BWB09 (*trpC+, ΔxynA ΔamyE*).

For the overproduction of XynA, we used plasmid pCS58, which was based on the multicopy expression plasmid pUB110, containing *xynA* cloned downstream of the strong constitutive *amyQ* promoter (21, 22). As a negative control, we constructed an empty plasmid based on pCS58 from which the *xynA* ORF was removed by cyclizing, using PCR primers BW34 and BW35 and self-ligation, resulting in plasmid pBW17.

## RNA extraction for RNA-seq

To examine the effect of xylanase overproduction on the transcriptome of *B. subtilis*, the BWB09 strains containing either pCS58 or pBW17 were grown in LB medium at 37°C for approximately 6 h when the cells entered the stationary phase and the cultures reached an $OD_{600}$ of approximately 4. The cultures were inoculated from overnight cultures, and 50 µg/mL of kanamycin was present in the medium to maintain the plasmids. The experiment was repeated one more time to provide a biological replicate.

RNA extraction was based on the methods described in (23, 24). Briefly, 2 mL of cells were collected by centrifugation [20,000 × relative centrifugal force (rcf)] at 4°C for 1 min. Cell pellets were resuspended in 0.4 mL of ice-cold growth medium and added to a screw cap tube containing 1.5 g of glass beads (0.1 mm), 0.4 mL of phenol chloroform/isoamyl alcohol (P/C/I) mixture (25:24:1), and 50 µL of 10% SDS, vortexed to mix, and flash frozen in liquid nitrogen. Cell disruption was achieved by bead beating (Precellys 24). After centrifugation, RNA in the upper aqueous phase was ethanol precipitated, washed twice with 70% ethanol, air dried, and dissolved in water. DNA was removed by DNase I (NEB) treatment. The pure total RNA was then extracted by a second round of P/C/I extraction, followed by ethanol precipitation and 70% ethanol washing, and finally dissolved in water and stored at −20°C.

## RNA-seq

Prior to deep sequencing, ribosomal RNA (rRNA) was removed using the MICROBExpress Bacterial mRNA Enrichment Kit (Thermo Fisher). Subsequently, the RNA-seq libraries were constructed using the NEBNext Ultra II Directional RNA Library Prep Kit for Illumina (New England Biolabs) and NEBNext Multiplex Oligos for Illumina (New England Biolabs), according to the manufacturer's protocols. Sequencing was performed on an Illumina NextSeq 550 System using NextSeq 500/550 High Output v2.5 kit (75-bp read length), and the raw data were processed using the web-based platform Galaxy. We aimed at a sequencing depth of around 7 million reads/library and eventually gained 7.28 million reads/library on average. *Trimmomatic* was used to trim adaptor sequence and filter bad reads (25). Trimmed reads were aligned to the *B. subtilis* reference genome (NC_000913) with *Bowtie2* (26). After mapping, aligned reads were counted by *FeatureCount* referred to in the BSU locus_tag in the annotation, detecting 4,406 genes, pseudogenes, and 178 RNAs (27). *Deseq2* was used to determine differentially expressed features between samples (28). The fold change, *P*-value, and functional description of all genes are listed in Table S3.

## GINtool

GINtool was written in C# as an Excel plugin (VSTO) and runs on Windows versions of Excel. A GINtool starter package with manual, data test files, and the GINtool execution file can be downloaded from GitHub (https://github.com/ScienceParkStudyGroup/GINtool/releases/tag/v.1.0.1.7).

## ACKNOWLEDGMENTS

We would like to thank Yoena Nossent, Niels Huiberts, Pascal Maas, and Anne van Winzum for coming up with the GINtool name, and Gertjan Kramer and Filipe Branco dos Santos for critical reading of the manuscript.

B.W. and M.B.M.J.K. were supported by an NWO-TTW (17833) grant awarded to J.L. and L.W.H.

## AUTHOR AFFILIATIONS

[1]Swammerdam Institute for Life Sciences, University of Amsterdam, Amsterdam, the Netherlands
[2]Molecular Microbiology, Amsterdam Institute of Molecular and Life Sciences, Vrije Universiteit Amsterdam, Amsterdam, the Netherlands

## AUTHOR ORCIDs

Biwen Wang http://orcid.org/0000-0003-4383-3902
Frans van der Kloet http://orcid.org/0000-0002-8573-2651
Mariah B. M. J. Kes http://orcid.org/0000-0002-4316-5783
Leendert W. Hamoen http://orcid.org/0000-0001-9251-1403

## FUNDING

| Funder | Grant(s) | Author(s) |
|---|---|---|
| Nederlandse Organisatie voor Wetenschappelijk Onderzoek (NWO) | 17833 | Joen Luirink |
| | | Leendert W. Hamoen |

## AUTHOR CONTRIBUTIONS

Biwen Wang, Data curation, Formal analysis, Investigation, Methodology, Validation, Visualization, Writing – original draft | Frans van der Kloet, Conceptualization, Formal analysis, Investigation, Methodology, Software, Validation, Visualization, Writing – original draft | Mariah B. M. J. Kes, Data curation, Formal analysis, Investigation, Methodology, Validation, Visualization, Writing – original draft | Joen Luirink, Formal analysis, Funding acquisition, Project administration, Supervision, Writing – review and editing | Leendert W. Hamoen, Conceptualization, Data curation, Formal analysis, Funding acquisition, Investigation, Methodology, Project administration, Supervision, Validation, Visualization, Writing – review and editing

## DATA AVAILABILITY

RNA-seq data have been submitted to and are accessible in the Gene Expression Omnibus (GEO), accession number GSE208571.

## ADDITIONAL FILES

The following material is available online.

## Supplemental Material

**Supplemental text (Spectrum03456-23-s0001.pdf).** GIN tool manual.

**Tables S1 and S2 (Spectrum03456-23-s0002.pdf).** Strains, plasmids, and primer sequences.

**Table S3 (Spectrum03456-23-s0003.xlsx).** Transcriptome data used for analyses.

## Open Peer Review

**PEER REVIEW HISTORY (review-history.pdf).** An accounting of the reviewer comments and feedback.

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
