## [Reviewer comments · Microbiology Spectrum]

Microbiology Spectrum

Improving Gene Set Enrichment Analysis (GSEA) by using regulation directionality

Biwen Wang, Frans Van der Kloet, Mariah Kes, Joen Luirink, and Leendert Hamoen

Corresponding Author(s): Leendert Hamoen, Universiteit van Amsterdam

Review Timeline:

Submission Date:	September 21, 2023
Editorial Decision:	November 4, 2023
Revision Received:	December 19, 2023
Accepted:	January 3, 2024

Editor: Tino Polen

Reviewer(s): The reviewers have opted to remain anonymous.

Transaction Report:

DOI: <https://doi.org/10.1128/spectrum.03456-23>

Re: Spectrum03456-23 (Improving Gene Set Enrichment Analysis (GSEA) by using regulation directionality)

Dear Professor Leendert Hamoen,

thank you for submitting your manuscript to Microbiology Spectrum.

I received expert reviewer comments on your manuscript suggesting minor modifications.

I do hope you find the comments below helpful and look forward to receiving a revised version from you.

Revision Guidelines

Sincerely,
Tino Polen
Editor
Microbiology Spectrum

Reviewer #1 (Comments for the Author):

GINtool is a very useful addition to gene set enrichment tools that are already available. It fills a hole where other tools do not look at positive or negative regulation, especially within the same regulon. The manuscript is already very well written and easy to follow. The tool is also easy to install and works as described in the manual.

Minor comments

Transcriptome data - the authors allude to that it can be used for other data analysis, but I think this could be made clearer in the abstract and introduction - e.g proteome data.

Line 39 genome rather than cells?

Line 54 - analysis platforms often only allow....

Line 76 - Specific gene

Page 8 - Description of how figure 2 is generate and what it actually shows needs to be made clearer. I read it three times and still can't quite see how you can work out up or down regulation?

Line 207 - doesn't read right

Line 230 - used

Line 261 - Lysogeny broth, not Luria-Bertani

Fig 1 - Orange not red and seeing differences in colour is difficult. Can this be improved?

Fig 2 - resD also marked in red, but not mentioned in legend?

Fig 3 - I thought that this was a clever presentation of data.

Point-by-point reply to reviewers' comments

We would like to thank the reviewer for the efforts and useful comments.

To clearly indicate our replies, we have written them in blue.

Reviewer #1:

GINtool is a very useful addition to gene set enrichment tools that are already available. It fills a hole where other tools do not look at positive or negative regulation, especially within the same regulon. The manuscript is already very well written and easy to follow. The tool is also easy to install and works as described in the manual.

We would like to thank the reviewer for the positive assessment

Minor comments

Transcriptome data - the authors allude to that it can be used for other data analysis, but I think this could be made clearer in the abstract and introduction - e.g proteome data.

We thank the reviewer for pointing this out. We have added the potential use for proteome data more clearly in the abstract (line 34 in the revision) and conclusion (line 257 in the revision).

Line 39 genome rather than cells?

Changed to genomes (line 40 in the revision).

Line 54 - analysis platforms often only allow...

We think that the original phrasing "However, due to the large number of genes measured, data interpretation requires focusing on a limited set of genes" is more accurate here (line 54 in the revision).

Line 76 - Specific gene

This suggestion has been added (line 76 in the revision).

Page 8 - Description of how figure 2 is generate and what it actually shows needs to be made clearer. I read it three times and still can't quite see how you can work out up or down regulation?

The description of how figure 2 is generated and what is shows has now been rewritten to make this more clear (lines 145-157 in the revision).

Line 207 - doesn't read right

We have rephrased the sentence (lines 209-211 in the revision).

Line 230 – used

Corrected

Line 261 - Lysogeny broth, not Luria-Bertani

Corrected.

Fig 1 - Orange not red and seeing differences in colour is difficult. Can this be improved?

We have changed the colour to blue to improve the visibility. Of note, users can easily customize their colour scheme in Excel to their own preferences.

Fig 2 - resD also marked in red, but not mentioned in legend?

Thank you for pointing this out. The marking of ResD is now included in the legend.

Fig 3 - I thought that this was a clever presentation of data.

We appreciate the compliment.

Re: Spectrum03456-23R1 (Improving Gene Set Enrichment Analysis (GSEA) by using regulation directionality)

Dear Professor Leendert Hamoen,

your revised manuscript has been accepted, and I am forwarding it to the ASM production staff for publication. Your paper will first be checked to make sure all elements meet the technical requirements. ASM staff will contact you if anything needs to be revised before copyediting and production can begin. Otherwise, you will be notified when your proofs are ready to be viewed.

Sincerely,
Tino Polen
Editor
Microbiology Spectrum